# Enabling long-distance hydrogen spillover in nonreducible metal-organic frameworks for catalytic reaction

Xiao-Jue Bai [1], Caoyu Yang[1,2] & Zhiyong Tang [1,2] ✉

Hydrogen spillover is an extraordinary effect in heterogeneous catalysis and hydrogen storage, which refers to the surface migration of metal particle-activated hydrogen atoms over the solid supports. Historical studies on this phenomenon have mostly been limited to reducible metal oxides where the long-distance proton-electron coupled migration mechanism has been established, yet the key question remains on how to surmount short-distance and defect-dependent hydrogen migration on nonreducible supports. By demerging hydrogen migration and hydrogenation reaction, here we demonstrate that the hydrogen spillover in nonreducible metal-organic frameworks (MOFs) can be finely modulated by the ligand functional groups or embedded water molecules, enabling significant long-distance (exceed 50 nm) movement of activated hydrogen. Furthermore, using sandwich nanostructured MOFs@Pt@MOFs as catalysts, we achieve highly selective hydrogenation of N-heteroarenes via controllable hydrogen spillover from Pt to MOFs-shell. We anticipate that this work will enhance the understanding of hydrogen spillover and shed light on de novo design of MOFs supported catalysts for many important reactions involving hydrogen.

The exploration of hydrogen spillover[1,2] not only accounts for many unconventional phenomena observed in hydrogenation or hydrogenolysis processes but also contributes to the recognition of the dynamic behavior of migrated hydrogen species on the catalyst surface[3], rendering the precise regulation of the reactions towards more efficient synthesis of chemicals[4-7]. It's commonly accepted that the movement of hydrogen atoms on the reducible metal oxide supports (e.g., $MoO_3$, $TiO_2$)[8,9] is thermodynamically feasible, ensuring long-distance migration. In this process, protons move over the support surface, and simultaneously electrons transfer through the conduction band of the material framework[3,10]. In contrast, the occurrence of hydrogen spillover on nonreducible metal oxide supports (e.g., aluminosilicates, $Al_2O_3$) has been debated for a long time[11-14] and only recently has it come to a consensus that the mobility of hydrogen atoms on nonreducible oxide supports is defect-dependent and confined to very short distances[15,16]. It should be noted that most industrial

catalysts are based on nonreducible supports because of their high thermochemical/structural stability and tunable acidity[11].

Beyond metal oxide supports, MOFs[17] formed via the coordination of metal ions with organic molecules have been recognized as a new generation of catalyst supports due to their porous crystalline structure, precise molecular/atomic arrangement[18], and synergistic catalysis effect[19,20]. Early studies on hydrogen spillover in MOFs focused on improving their hydrogen adsorption capability by doping metal catalysts[21,22]. Typically, hydrogen spillover on carboxylate-based MOFs (e.g., MOF-5) has been found to be a short-range hydrogenation process of benzene-carboxylate and thermodynamically blocked[23], while metal vacancy defects are critical for eliminating high-energy barriers for hydrogen migration[24]. Subsequently, thermoanalysis technology has been used to quantify the migration distance of hydrogen in zeolitic imidazolate framework-8 (ZIF-8)[25], where hydrogen spillover displays a limited spatial scope (i.e., in the vicinity of

[1]Chinese Academy of Sciences (CAS) Key Laboratory of Nanosystem and Hierarchy Fabrication, National Center for Nanoscience and Technology, Beijing, PR China. [2]University of Chinese Academy of Sciences, Beijing, PR China. ✉e-mail: zytang@nanoctr.cn

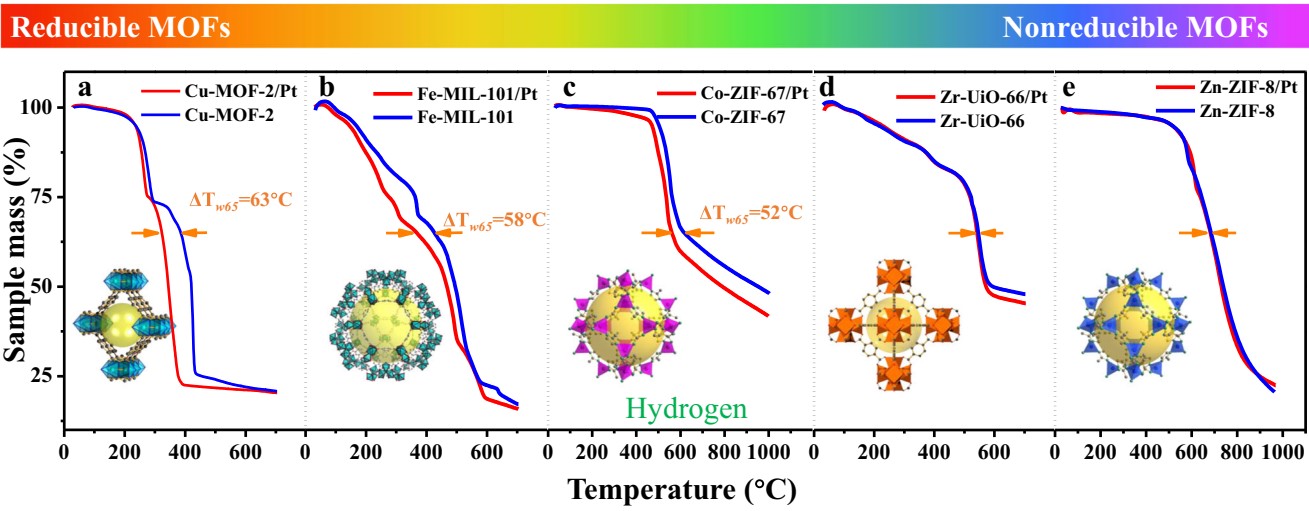

**Fig. 1 | Thermal stability of several typical MOFs and MOFs/Pt in flowing H₂ gas.** TGA curve and $\Delta T_{w65}$ of MOFs and MOFs/Pt: **a** Cu-MOF-2 system. **b** Fe-MIL-101 system. **c** Co-ZIF-67 system. **d** Zr-UiO-66 system. **e** Zn-ZIF-8 system.

metal particles) even at elevated temperature, similar to nonreducible metal oxides[26]. Very recently, a study claimed a long-region hydrogen spillover path in the Pt@MOF-801 system via water assistance, but unfortunately, the accompanying structural collapse of MOFs made the spillover mechanism of activated hydrogen even more puzzling and the real migration distance hardly estimated[27]. At present, much remains unknown about promoting or restraining hydrogen spillover inside MOFs, let alone regulating the migration distances and subsequent catalytic reactions.

Here, we first establish the mechanism of hydrogen spillover in reducible MOFs by hydrogen pyrolysis technology, which gives rise to metal node reduction and inevitable framework damage. Then, by decoupling hydrogen migration and hydrogenation reaction, we demonstrate that the hydrogen spillover in nonreducible MOFs can be finely modulated by the ligand functional groups or embedded water molecules, endowing significant long-distance migration or negligible migration of activated hydrogen while keeping structural stability. Furthermore, using sandwich nanostructured MOFs@Pt@MOFs as catalysts, we achieve highly selective hydrogenation of N-heteroarenes via controllable hydrogen spillover from Pt to MOFs-shell.

## Results

### Reduction behavior of MOFs in hydrogen spillover

In order to deeply understand the hydrogen spillover behavior of varied MOFs including reducible and nonreducible ones, we first examined the thermal stability of frequently-used MOFs with different reduction potential energy[28] (Supplementary Fig. 1) (Zn-based ZIF-8, Zr-based UiO-66, Co-based ZIF-67, Fe-based MIL-101 and Cu-based MOF-2, Supplementary Figs. 2–6) and their composite with immobilized Pt nanoparticles (denoted as MOFs/Pt, Supplementary Figs. 7–11) in flowing H₂ gas by thermogravimetric analysis (TGA).

As shown by the TGA curves (Fig. 1a–c), MOFs/Pt with high reduction potential energy (Cu-based, Fe-based, and Co-based ones) are more readily decomposed than the pristine MOFs. Using Cu-based MOFs as an example, the temperature at which the weight fraction w decreases to 65 wt% ($T_{w65}$) for Cu-MOF-2 and Cu-MOF-2/Pt in H₂ flows is 386 °C and 323 °C, respectively. The evident difference of $\Delta T_{w65}$ is caused by distinct decomposition mechanisms. In the Cu-MOF-2/Pt system, hydrogen atoms are produced by the dissociation of gaseous H₂ molecules on the embedded Pt surface, which subsequently migrate from Pt to Cu-MOF-2 via hydrogen spillover. As the split hydrogen atoms have higher reactivity than H₂, hydrogenolysis of Cu-MOF-2 happens more vigorously, where Cu-O coordination bonds are

cleaved and Cu²⁺ species are reduced at lower temperatures. The reduction of Cu²⁺ ions is proved by X-ray diffraction (XRD) and X-ray photoelectron spectroscopy (XPS) (Supplementary Figs. 12 and 13). A similar phenomenon is observed in the Fe-based and Co-based systems (Supplementary Figs. 14 and 15).

Conversely, Fig. 1d, e indicates that such hydrogen spillover-promoted decomposition does not take effect in the Zr-UiO-66/Pt and Zn-ZIF-8/Pt systems. This is likely a consequence of their much lower reduction potential energy, meaning that the migrated hydrogen atoms cannot reduce Zr⁴⁺ and Zn²⁺ ions, and thus the frameworks of Zr-UiO-66/Pt and Zn-ZIF-8/Pt remain stable against hydrogenolysis (Supplementary Figs. 16–19).

To sum up, analogously to all the reported results[25,27,29], the inherent crystal structure of the reducible MOFs is destroyed by activated hydrogen, which greatly reduces their practical value as the catalytic supports. Therefore, we focus the study on the hydrogen spillover effect of more meaningful and stable nonreducible MOFs. Zn-based ZIF-8 is chosen as the platform material considering its high thermal stability (Supplementary Figs. 20–24), functional group flexibility (Supplementary Fig. 25), and extremely low reduction potential energy (Supplementary Fig. 1).

### Characterization of sandwich model catalyst

The sandwich nanostructured MOFs@Pt@MOFs catalyst[19] with the outer MOFs-shell of varied thickness and functional groups serves as a perfect model for exploring hydrogen spillover, ensuring the identical diffusion pathways of hydrogen atoms starting from Pt to the different external surfaces on MOFs. Accordingly, highly symmetrical and uniform Zn-ZIF-8@Pt@Zn-ZIF-8 nanocubes were prepared by electrostatic adsorption of pre-synthesized 3 nm Pt nanoparticles onto 100 nm Zn-ZIF-8 nanocube cores[30] (Supplementary Fig. 26) followed by liquid-phase epitaxial growth of another Zn-ZIF-8 shells (Supplementary Fig. 27). Note that Zn-ZIF-8 is composed of Zn²⁺ and 2-methylimidazole. Characterizations of the morphology and structure clearly validate the successful synthesis of sandwich Zn-ZIF-8@Pt@Zn-ZIF-8 nanocubes (Supplementary Figs. 28–32). In addition, XPS depth profiles show that the original Zn-ZIF-8@Pt@Zn-ZIF-8 possesses only C, N, O, and Zn elements whereas the Pt element is discerned after argon plasma etching, suggesting that Pt nanoparticles are completely encapsulated by the outer Zn-ZIF-8 shell (Supplementary Fig. 33).

The thickness of the outer Zn-ZIF-8 layer is then adjusted from 15 to 50 nm, in order to make it as a ruler for precisely measuring the

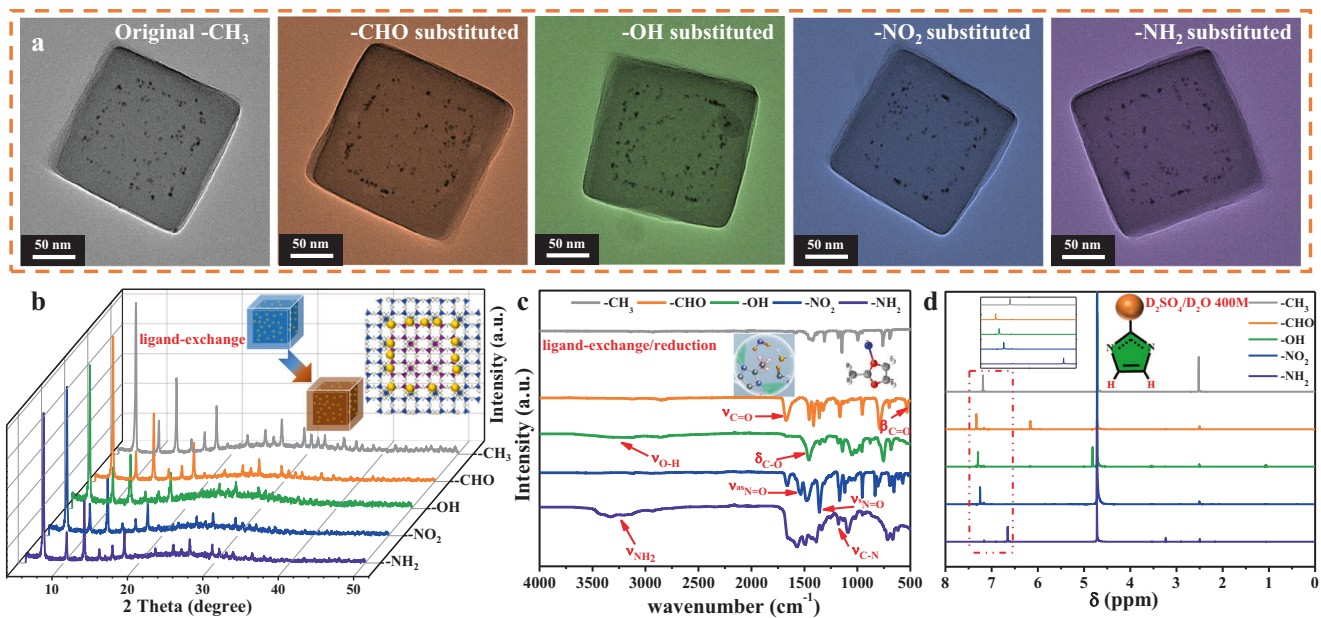

**Fig. 2 | Solvent-assisted ligand exchange/reduction process for Zn-ZIF-8@Pt@Zn-ZIF-8. a** TEM images of Zn-ZIFs@Pt@Zn-ZIFs homologs (scale bar: 50 nm). **b** XRD patterns. **c** ATR-FTIR spectra and **d** ¹H NMR spectra of Zn-ZIFs@Pt@Zn-ZIFs homologs after ligand exchange.

spillover distance of hydrogen atoms (Supplementary Fig. 34). Note that the sample is synthesized in water, so the water molecules are unavoidably adsorbed in Zn-ZIFs pores if no special activation treatment is performed (denoted as Zn-ZIF-8@Pt@Zn-ZIF-8 (H₂O)). In order to further obtain Zn-ZIFs@Pt@Zn-ZIFs homologs, solvent-assisted ligand exchange/reduction strategy[31,32] was adopted to replace 2-methyl group in Zn-ZIF-8 with different functional groups including CHO, OH, NO₂ and NH₂ (Supplementary Fig. 35), denoted as Zn-ZIFs (CHO), Zn-ZIFs (OH), Zn-ZIFs (NO₂) and Zn-ZIFs (NH₂), respectively. As manifested in Fig. 2a, b, the morphology and crystal structure of homologs remain unchanged after the ligand exchange/reduction process. Figure 2c presents the attenuated total reflectance Fourier transform infrared (ATR-FTIR) spectra of Zn-ZIF-8@Pt@Zn-ZIF-8 before and after ligand exchange/reduction. The incorporations of new functional groups are confirmed by appearance of $\nu_{C=O}$ at 1700 cm⁻¹ and $\beta_{C=O}$ at 531 cm⁻¹ (CHO), $\delta_{C-O}$ at 1459 cm⁻¹ and $\nu_{O-H}$ at round 3310 cm⁻¹ (OH), $\nu_{as(N-O)}$ at 1549 cm⁻¹ and $\nu_{s(N-O)}$ at 1359 cm⁻¹ (NO₂), $\nu_{C-N}$ at 1178 cm⁻¹ and $\nu_{N-H}$ at the range of 3100–3500 cm⁻¹ (NH₂), respectively. The H-nuclear magnetic resonance (¹H NMR) spectra quantitatively reveal that the ligand replacement ratio of all homologs exceeds 85% (Fig. 2d).

## Evaluation of hydrogen spillover efficiency

Zn-ZIF-8 is characteristic of a very small aperture window (0.34 nm) (Supplementary Fig. 36), allowing selective diffusion of H₂ (0.29 nm) rather than large-sized molecules like cyclooctene (0.6 nm) (Supplementary Fig. 37). The adsorption experiments prove its size-sieving effect (Supplementary Fig. 38), guaranteeing that the reaction between hydrogen atom and organic reactant occurs only on the external surface of the MOFs matrix (Supplementary Figs. 39 and 40). Therefore, catalytic hydrogenation of cyclooctene is selected to evaluate the hydrogen spillover behavior of Zn-ZIFs@Pt@Zn-ZIFs homologs with different functional groups and shell thicknesses (Fig. 3a, Supplementary Figs. 34 and 41–44).

First, we explore the effect of functional groups on spillover hydrogenation while controlling the shell thickness of all the samples to be 15 nm. As shown in Supplementary Table S1 and Fig. 3b, Zn-ZIF-8@Pt@Zn-ZIF-8 with water removal exhibits very low catalytic activity towards cyclooctene hydrogenation even under the elevated reaction

conditions (gray curve in Fig. 3b, Supplementary Fig. 45), demonstrating that the hydrogen spillover is negligible in pure Zn-ZIF-8 below 100 °C. Our result is consistent with earlier reports[33,34] that Pt@ZIF-8 is inactive for the hydrogenation of organic molecules. In sharp contrast, Zn-ZIF-8@Pt@Zn-ZIF-8 (H₂O) without water removal shows unexpectedly high catalytic activity towards cyclooctene hydrogenation (red curve in Fig. 3b). These results clearly indicate that water plays a crucial role in the migration of activated hydrogen. For other homologs with water removal, under the same condition, the hydrogenation can take place on Zn-ZIFs@Pt@Zn-ZIFs (OH), Zn-ZIFs@Pt@Zn-ZIFs (CHO) and Zn-ZIFs@Pt@Zn-ZIFs (NH₂) (Fig. 3b), manifesting the occurrence of hydrogen spillover from Pt nanoparticles to MOFs surface via these H acceptors. Among these homologs, the highest activity of Zn-ZIF-8@Pt@Zn-ZIF-8 (H₂O) is attributed to the high mobility of molecular water. Furthermore, the recovered Zn-ZIF-8@Pt@Zn-ZIF-8 (H₂O) catalyst of stable structure (Supplementary Fig. 46) is rapidly deactivated due to the loss of water, whereas the other homologs maintain high catalytic activity, validating the major role of water in hydrogen migration (Fig. 3c and Supplementary Table S2).

Next, we change the shell thicknesses of the above four homologs including Zn-ZIF-8@Pt@Zn-ZIF-8 (H₂O), Zn-ZIFs@Pt@Zn-ZIFs (OH), Zn-ZIFs@Pt@Zn-ZIFs (CHO) and Zn-ZIFs@Pt@Zn-ZIFs (NH₂). Here, cyclooctene hydrogenation conversion/100 is defined as the spillover intensity. The spillover intensity generally increases with thinner shell thickness, higher reaction temperature, and longer reaction time (Supplementary Fig. 47 and Supplementary Tables S3 and S4). At the specific temperature and time, the spillover intensity fits well to an equation: $y = ax + b$, where y refers to the spillover intensity, $a$ represents the spillover decay factor, $x$ stands for the shell thickness, and $b$ is the maximum spillover intensity close to 1 (Fig. 3d and Supplementary Fig. 48). Noteworthily, Zn-ZIF-8@Pt@Zn-ZIF-8 (H₂O) exhibits the smallest spillover decay factor of 0.01997 at 80 °C and 80-min reaction, and its hydrogen spillover even exceeds 50 nm (red line in Fig. 3d), representing an amazing distance that has been never found on nonreducible oxides. In addition, the sandwich catalysts with large-sized Pt NPs (ca. 5 nm) exhibit nearly the same spillover decay factor as the above-used one but lower hydrogenation activity, meaning that particle size only affects hydrogen activation independent of spillover intensity (Supplementary Figs. 49–51). Furthermore, the catalytic

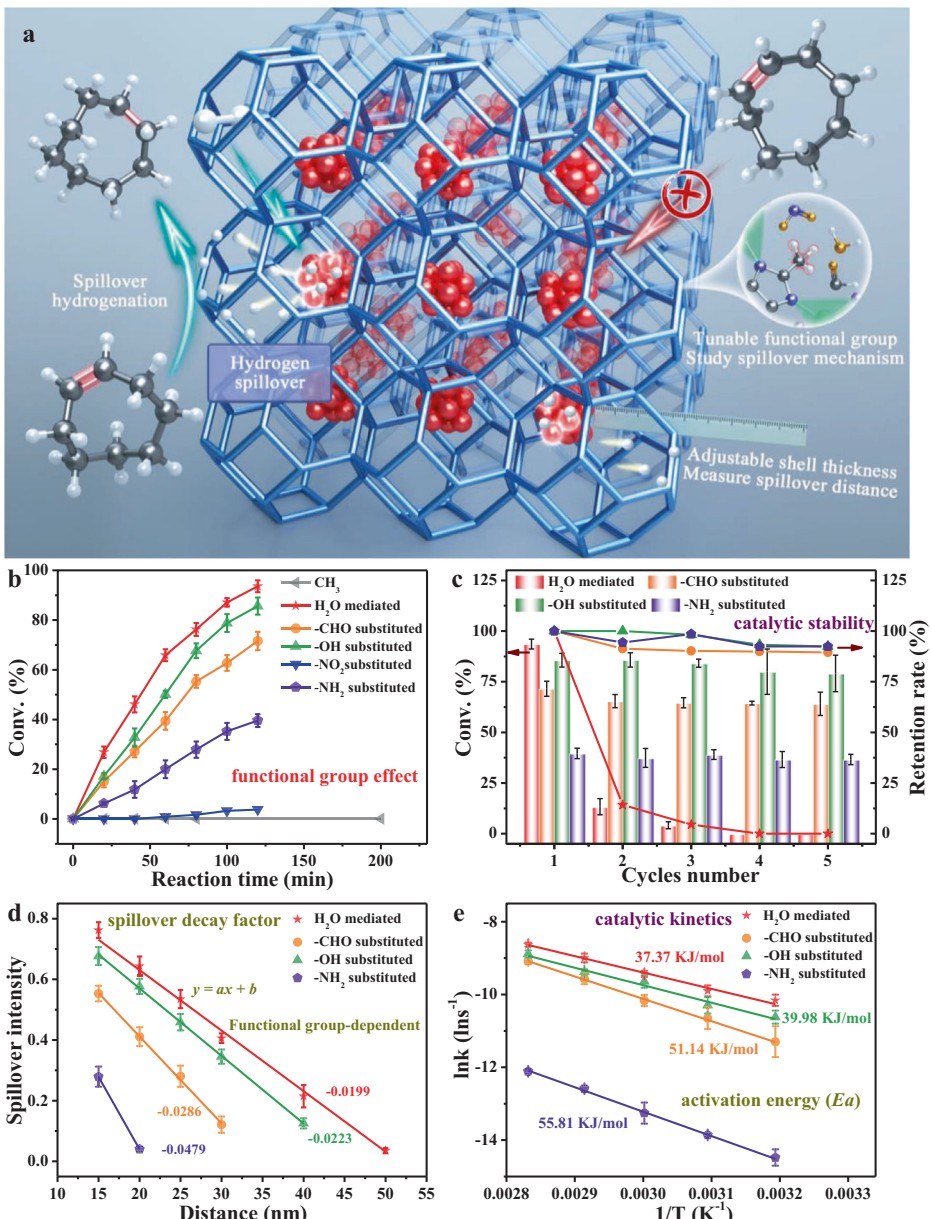

**Fig. 3 | Spillover hydrogenation of cyclooctene. a** Hydrogenation pathway of cyclooctene on Zn-ZIFs@Pt@Zn-ZIFs. **b** Catalytic conversion of cyclooctene by Zn-ZIFs@Pt@Zn-ZIFs homologs. **c** Catalytic stability of Zn-ZIFs@Pt@Zn-ZIFs homologs. **d** Linear relationship between spillover intensity and spillover distance of Zn-ZIFs@Pt@Zn-ZIFs homologs (80-min reaction). **e** Plot of ln$k$ as a function of (1/$T$) for various catalysts. All reactions were performed using catalysts with the same amount of Pt NPs. All the reactions are carried out under a shell thickness of 15 nm, temperature of 80 °C, and time of 120 min if not specially mentioned. (error bar: standard deviation).

kinetics is studied. As shown in Supplementary Figs. 52–55, all four homolog catalysts display the zero-reaction order characteristic in the initial period, enabling us to quantitively extract the activation energy ($E_a$) using the Arrhenius equation: $k = Ae^{-Ea/RT}$. Accordingly, $E_a$ is determined to be 37.37 kJ/mol for Zn-ZIFs@Pt@Zn-ZIFs (H$_2$O), 39.98 kJ/mol for Zn-ZIFs@Pt@Zn-ZIFs (OH), 51.14 kJ/mol for Zn-ZIFs@Pt@Zn-ZIFs (CHO) and 55.81 kJ/mol for Zn-ZIFs@Pt@Zn-ZIFs (NH$_2$) (Fig. 3e and Supplementary Fig. 56). Such low $E_a$ suggests the kinetically favorable pathway of Zn-ZIFs@Pt@Zn-ZIFs in the catalytic hydrogenation through hydrogen spillover at low temperatures.

In order to elucidate the process of hydrogen spillover, isotope deuterium labeling experiments were adopted to track the trajectories of water and hydrogen by detecting the probe molecule (cyclooctene) mass spectrometry (Supplementary Fig. 57). As shown in Supplementary Fig. 58, the hydrogenation and deuteration of

cyclooctene are hardly detected in Zn-ZIF-8(D$_2$O)-D$_2$ system, indicating that H-D exchange on probe cyclooctene is difficult to occur under the mild condition. At the same time, complete spillover hydrogenation and spillover deuteration can be clearly observed in Zn-ZIF-8@Pt@Zn-ZIF-8 (H$_2$O)-H$_2$ system and Zn-ZIF-8@Pt@Zn-ZIF-8 (D$_2$O)-D$_2$ system, respectively. Interestingly, whether in the Zn-ZIF-8@Pt@Zn-ZIF-8 (D$_2$O)-H$_2$ system or in the Zn-ZIF-8@Pt@Zn-ZIF-8 (H$_2$O)-D$_2$ system, cyclooctene is hydrogenated to cyclooctane, [D$_1$]-cyclooctane and [D$_2$]-cyclooctane. The above analysis demonstrates that H$_2$ splitting occurs on Pt NPs, and the activated hydrogen atoms diffuse across the MOF structure by the water-assist path accompanied by exchange with water. In addition, as for the Zn-ZIFs@Pt@Zn-ZIFs (CHO)-D$_2$ system, the product of [D$_2$]-cyclooctane clarifies the migration of D atoms across MOFs containing CHO functional groups.

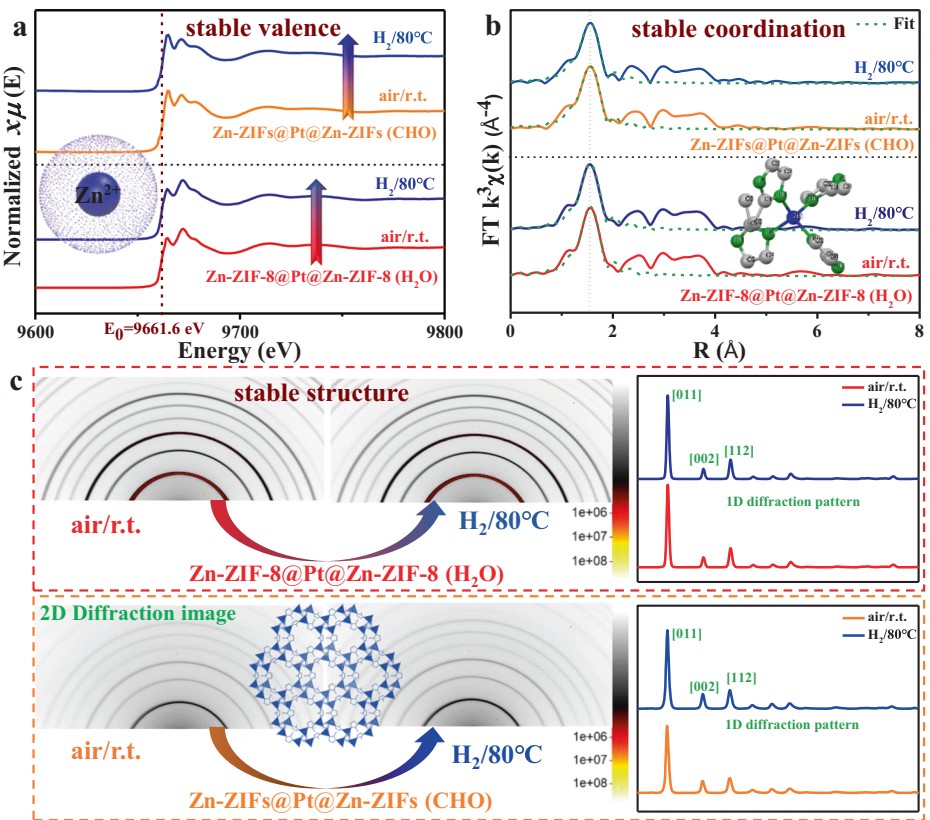

**Fig. 4 | Effect of metal nodes on hydrogen spillover: in situ XAS-XRD characterization of Zn-ZIF-8@Pt@Zn-ZIF-8 (H₂O) and Zn-ZIFs@Pt@Zn-ZIFs (CHO). a** Zn K-edge XANES and **b** Zn K-edge EXAFS spectra in R-space collected on as-prepared Zn-ZIFs@Pt@Zn-ZIFs under air at room temperature or hydrogen at 80 °C. **c**, XRD patterns of as-prepared Zn-ZIFs@Pt@Zn-ZIFs under air at room temperature or hydrogen at 80 °C.

## Visual evidence of hydrogen spillover

The color change of WO₃ is a classic visual method for assessing the hydrogen spillover effect[35]. Upon mixing WO₃ with varied Zn-ZIFs@Pt@Zn-ZIFs homologs, color-change experiments further confirm the distance and functional group-dependent hydrogen spillover in Zn-ZIFs (Supplementary Figs. 59 and 60).

## Exploration of hydrogen spillover mechanism

To decipher the spillover mechanism in Zn-ZIFs, we selected two representative catalysts, H₂O-mediated Zn-ZIF-8@Pt@Zn-ZIF-8 (H₂O) and functional group-mediated Zn-ZIFs@Pt@Zn-ZIFs (CHO). The coordination nature of metal atoms was investigated by in situ X-ray absorption spectroscopy (XAS)-XRD combined spectroscopy. Clearly, both Zn K-edge X-ray absorption near-edge structure (XANES) and extended X-ray absorption fine structure (EXAFS) spectra of Zn-ZIF-8@Pt@Zn-ZIF-8 (H₂O) and Zn-ZIFs@Pt@Zn-ZIFs (CHO) barely change under hot hydrogen atmosphere (Fig. 4a, b, Supplementary Figs. 61–66 and Supplementary Table S5), indicating that hydrogen spillover in Zn-ZIFs does not change either the oxidation state or coordination environment of Zn atoms. In addition, the corresponding XRD patterns show that both samples maintain high structural stability during the whole H₂-heat process (Fig. 4c and Supplementary Fig. 66).

To further explore the H-binding sites in nonreducible MOFs, we employed in situ XPS technology to detect the valence of surface elements (Supplementary Figs. 67–70). The Zn *2p* and N *1 s* peaks of as-prepared Zn-ZIF-8@Pt@Zn-ZIF-8 (H₂O) are located at 1021/1044 eV and 389.8 eV, respectively, and these peaks remain almost constant under hydrogen atmosphere. Note that the slight shift of these peaks is likely caused by the introduction of H (or electron) in the system that affects the Zn-N bond. As for Zn-ZIFs@Pt@Zn-ZIFs (CHO), its Zn *2p*

peaks also remain unchanged during the entire in situ test, whereas the corresponding O *1s* peaks clearly shift to higher energy (Supplementary Fig. 70), indicating that the O species of the aldehyde group is the binding site of the H atom. Altogether, Zn species retain their original valence state after contact with H₂ even at elevated temperatures, and hydrogen spillover occurs on these nonreducible MOFs and does not involve the metal redox process. Therefore, we suppose that the oxygen site of the aldehyde group acts as a binding site for the hydrogen atom, which is then transferred to the oxygen site of the next aldehyde group. For the Zn-ZIF-8@Pt@Zn-ZIF-8 (H₂O) system, the embedded water acts as the functional group to promote hydrogen spillover. As proved by Supplementary Fig. 71, the water-containing catalyst delivers high catalytic activity for cyclooctene hydrogenation, whereas the water-free catalyst is inactive.

## Theoretical calculation of hydrogen migration

To probe the mechanism of hydrogen adsorption and migration on Zn-ZIFs supports, first-principles atomistic simulation calculations were carried out. As shown in Supplementary Fig. 72, 1.61 eV is required to transfer a dissociated hydrogen atom among the carbon sites of Zn-ZIF-8 (C1 → C2), which is energetically improbable at ambient temperature[25]. Furthermore, this calculated migration energy of the dissociated hydrogen atom on Zn-ZIF-8 is only slightly influenced by the presence of water in the pores. Regardless of whether hydrogen atoms migrate in the form of H₃O• or in the water environment (VASPsol), the energy barriers are both about 1.5 eV (Supplementary Figs. 73 and 74). Hence, we deduce that electrons migrate along the framework of Zn-ZIF-8 simultaneously with water-assisted proton hopping, which is affected by the number of water molecules. Along the water molecular chain, the energy barrier of proton migration is only 0.2 eV (Fig. 5a); while this value reaches 0.66 eV when involved

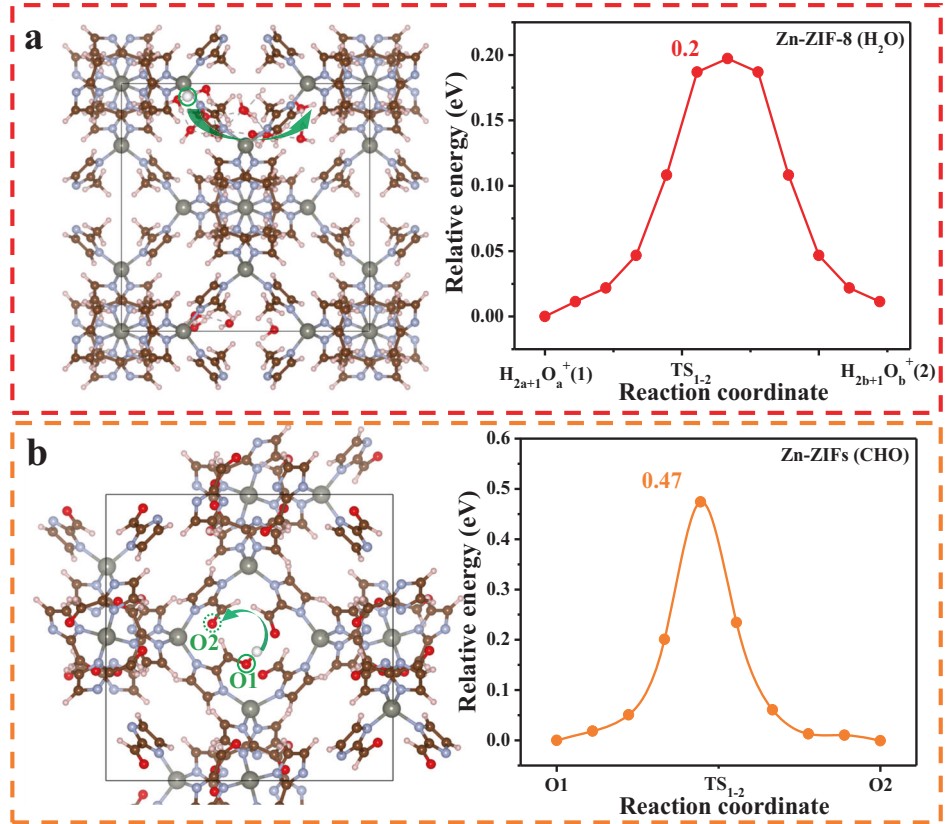

**Fig. 5 | Mechanism of hydrogen migration on Zn-ZIF-8 ($H_2O$) and Zn-ZIFs (CHO). The method used is the nudged elastic band (NEB). a** Hydrogen spillover (activation energy $E_{act}$) on Zn-ZIF-8 ($H_2O$) (water-assisted proton hopping of left model): $E_{act}$ = 0.2 eV. **b** Hydrogen spillover on aldehyde-Zn-ZIFs (steps O1-TS$_{1\text{-}2}$-O2 of left model): $E_{act}$ = 0.47 eV.

with two water molecules (i.e., $H_5O_2^+$) (Supplementary Fig. 75). This calculation explains the reduced catalytic activity of the recovered sample (Fig. 3c).

In regard to aldehyde-Zn-ZIFs, dissociative hydrogen is found to adsorb on the O site (Supplementary Fig. 70). The transfer of hydrogen species between neighboring oxygen sites has an activation energy barrier of 0.47 eV (Fig. 5b). In comparison, for nitro-ZIFs, although H also migrates through the O··H··O pathway, its migration barrier is as high as 0.86 eV (Supplementary Fig. 76), which explains its low hydrogenation activity (Fig. 3b).

### Catalytic application of controlled spillover hydrogenation
Finally, the hydrogen spillover effect in MOFs is applied to regulate practical catalytic reactions. Partial hydrogenation products of halogen-substituted N-heteroarenes are known to be core structural motifs in both fine and bulk chemicals, while the strong adsorption between substrate and metal catalyst often leads to over-hydrogenation and dehalogenation[36]. We choose 5-chloroquinoline as a substrate based on the fact that its large size prevents diffusion through the pores, so the hydrogenation is completely derived from spillover hydrogen (Fig. 6a). Impressively, the Zn-ZIFs@Pt@Zn-ZIFs (-CHO and $H_2O$) catalysts not only deliver a high catalytic activity similar to the commercial Pt/C, reducible TiO$_2$-supported Pt and nonreducible Al$_2$O$_3$-supported Pt catalysts, but also endow an unprecedentedly high selectivity of >99% to the primary product of 5-chloro-1,2,3,4-tetrahydroquinoline under the same reaction condition (Fig. 6b, Supplementary Figs. 77–79 and Supplementary Table S6). Noteworthily, this result is comparable to the best-reported heterogeneous catalysts (Supplementary Table S7). In addition, with the increase of shell thickness, the conversion rate of 5-chloroquine gradually decreases while the selectivity remains stable (Fig. 6c and

Supplementary Figs. 80–83), which is consistent with the concept of spillover distance decay factor (Fig. 3d).

## Discussion
In summary, our work clearly documents the process of hydrogen spillover in reducible MOFs like Cu-MOF-2, which gives rise to metal node reduction and inevitable framework damage. Similar to the conventional nonreducible metal oxides, hydrogen spillover in the nonreducible Zn-ZIF-8 is inefficient, but distinctively it can be largely promoted by the ligand functional groups or embedded water molecules, endowing significant long-distance migration of activated hydrogen while keeping structural stability. Looking forward, decoupling hydrogen activation and hydrogenation reaction by leveraging the hydrogen spillover effect in MOFs will bring great opportunities to implement many challenging reactions that require long regions and high stereoselectivity, especially benefitting from the rich designability of MOFs.

## Methods
### Chemicals
Chloroplatinic acid (H$_2$PtCl$_6$, 37 wt% Pt), zinc nitrate hexahydrate (Zn(NO3)2·6H2O), zirconium chloride (ZrCl$_4$), cobalt nitrate hexahydrate (Co(NO3)2·6H2O), ferric chloride hexahydrate (FeCl$_3$·6H2O), copper nitrate trihydrate (Cu(NO3)2·3H2O), terephthalic acid (H2BDC), 2-methylimidazole, imidazole-2-carboxaldehyde, 2-nitroimidazole, 2-aminoimidazole and sodium borohydride (NaBH4) were bought from Sinopharm Chemical Reagent Co., Ltd. Polyvinylpyrrolidone (PVP, Mw = 55,000) and hexadecyl trimethyl ammonium bromide (CTAB) were obtained from Sigma-Aldrich. Acetic acid, deuterosulfuric acid, 1-hexene, cyclooctene, and 5-chloroquine were purchased from Beijing InnoChem Science & Technology Co., Ltd.

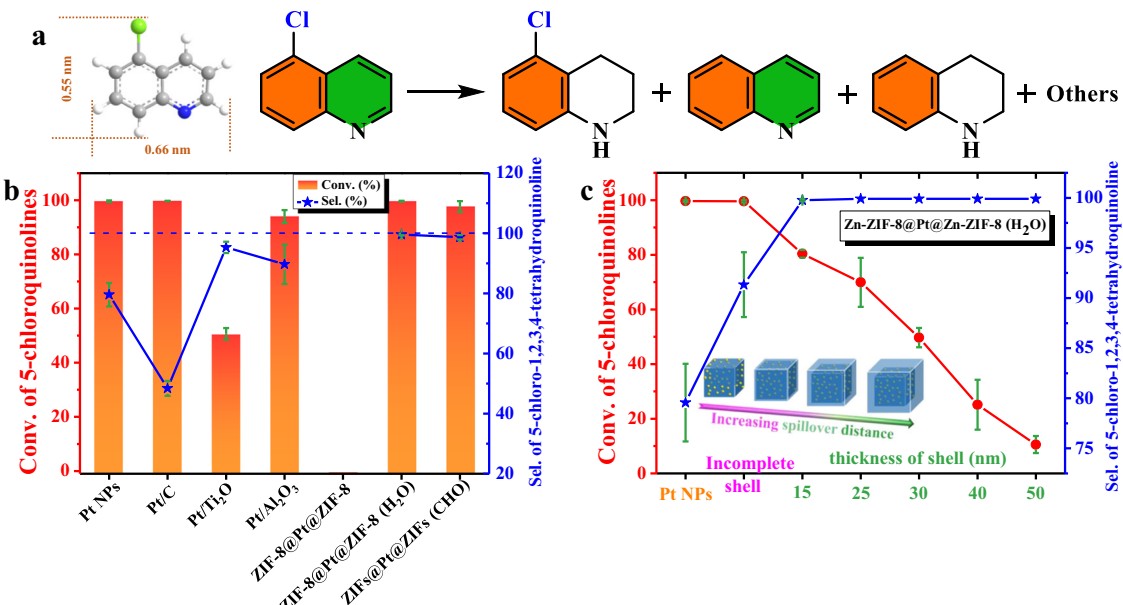

**Fig. 6 | Catalytic performance of Zn-ZIFs@Pt@Zn-ZIFs catalysts. a** Scheme of spillover hydrogenation of 5-chloroquinolines. **b** Conversion ratio and selectivity of 5-chloroquinolines catalyzed by various catalysts. **c** Conversion ratio and selectivity of 5-chloroquinolines catalyzed by Zn-ZIF-8@Pt@Zn-ZIF-8 (H₂O) with different shell thickness. Note: the reactions are carried out under the time of 240 min in group b and 200 min in group c. All reactions were performed using catalysts with the same amount of Pt NPs. (error bar: standard deviation).

Other analytical grade solvents including methanol (CH3OH), ethanol (C2H5OH), isopropanol (C3H7OH), and N, N-dimethylformamide (DMF) were supplied by Sinopharm. Carbon-supported Pt (Pt/C, 20 wt% Pt) was bought from Johnson Matthey Corp. WO₃ was obtained from Energy Chemical (Shanghai). The water used in this work was deionized water. All the chemicals were used without further purification.

### Synthesis of Pt nanoparticles (NPs)

Pt NPs (ca. 3 nm) were synthesized according to the literature with minor modification[19,37]. In a typical procedure, 33.2 mg PVP (Mw = 55,000) was dissolved in 48.8 mL ethanol, and then 1.2 mL H₂PtCl₆ aqueous solution (50 mM) was added drop by drop. After stirring for about 2 min at room temperature, the solution was heated at 83 °C for 4 h under air to synthesize the PVP-stabilized Pt NPs. The concentration of as-synthesized Pt NPs solution was about 1.2 mM and used directly without further treatment.

Pt NPs (ca. 5 nm) were synthesized by a second growth of PVP-Pt NPs. In a typical procedure, 25 mL of 1.2 mM Pt NPs (ca. 3 nm) solution was dispersed in 24.4 mL ethanol, and then 0.6 mL H₂PtCl₆ aqueous solution (50 mM) was added drop by drop. After stirring for about 2 min at room temperature, the solution was heated at 83 °C for 3 h under air to synthesize the PVP-stabilized Pt NPs. The concentration of as-synthesized Pt NPs solution was about 1.2 mM and used directly without further treatment.

### General synthetic method for MOFs/Pt

All five MOFs used in this study, Cu-MOF-2, Fe-MIL-101, Co-ZIF-67, Zr-UiO-66, and Zn-ZIF-8, were synthesized according to previous studies[38–40] with slight modification to optimize the crystallinity (Supplementary Information). The resulting powder was collected by centrifugation and further dried under vacuum at 80 °C for 12 h. 10 mg MOFs was redispersed in 4 mL ethanol, and then 2 mL Pt NPs solution (1.2 mM) was added drop by drop under stirring. Afterward, the solution was left stirred for 2 h at room temperature. As-synthesized MOFs/Pt were centrifuged at 4800 × g for 3 min and dried at 80 °C in a vacuum.

### Synthesis of cubic Zn-ZIF-8

3.5 mL[30] aqueous solution containing 790 mM 2-methylimidazole and 0.55 mM CTAB was stirred at 500 rpm for 5 min, followed by the addition of 0.5 mL 97.5 mM Zn(NO₃)₂·6H₂O aqueous solution. The mixture was stirred for another 5 min at 500 rpm. Then, the solution was left undisturbed for 3 h at room temperature. As-synthesized Zn-ZIF-8 nanocubes were centrifuged at 7800 × g for 10 min and redispersed in 3 mL ethanol.

### Synthesis of cubic Zn-ZIF-8@Pt

3 mL ZIF-8 ethanol solution was ultrasonicated for 10 min, and then 3 mL Pt NPs solution (1.2 mM) was added drop by drop under stirring. Then, the solution was left stirred for 1 h at room temperature. As-synthesized Zn-ZIF-8@Pt was centrifuged at 7800 × g for 8 min and redispersed in 0.9 mL water (marked as core source).

### Synthesis of cubic Zn-ZIF-8@Pt@Zn-ZIF-8

Take the Zn-ZIF-8@Pt@Zn-ZIF-8 (15 nm) as an example. 0.3 mL Zn-ZIF-8@Pt aqueous solution (core source) was dispersed in 3.5 mL aqueous solution containing 790 mM 2-methylimidazole and 0.55 mM CTAB, followed by the addition of 0.5 mL 97.5 mM Zn(NO₃)₂·6H₂O aqueous solution. The mixture was stirred for another 5 min at 500 rpm. Then, the solution was left undisturbed for 3 h at room temperature. The synthesized cubic Zn-ZIF-8@Pt@Zn-ZIF-8 (15 nm) were centrifuged at 6000 × g for 5 min.

For the synthesis of Zn-ZIF-8@Pt@Zn-ZIF-8 (20 nm, 25 nm, 30 nm, and incomplete shell), the concentration of precursors solution (Zn(NO₃)₂·6H₂O and 2-methylimidazole) was adjusted, while other parameters were held to identical values as Zn-ZIF-8@Pt@Zn-ZIF-8 (15 nm) synthesis. The concentration of 2-methylimidazole was 316 mM, 474 mM, 632 mM, and 112.8 mM, and the concentration of Zn(NO₃)₂·6H₂O was 39 mM, 58.5 mM, 778 mM, and 13.6 mM, corresponding to the shell thickness of 20 nm, 25 nm, 30 nm, and incomplete shell, respectively.

For the synthesis of Zn-ZIF-8@Pt@Zn-ZIF-8 (40 nm and 50 nm), the above Zn-ZIF-8@Pt@Zn-ZIF-8 (25 nm and 30 nm) were used as the core source, respectively. The concentration of 2-methylimidazole was

564 mM and 790 mM, and the concentration of $Zn(NO_3)_2 \cdot 6H_2O$ was 68.3 mM and 97.5 mM, corresponding to the thickness of 40 nm and 50 nm, respectively. Other parameters were held to identical values as Zn-ZIF-8@Pt@Zn-ZIF-8 (15 nm) synthesis.

## Synthesis of cubic Zn-ZIFs@Pt@Zn-ZIFs homolog

An as-prepared Zn-ZIF-8@Pt@Zn-ZIF-8 was solvothermally treated with a new linker (2-nitroimidazole (nIm), 2-aminoimidazole (aIm) and imidazole-2-carboxaldehyde (Ica)) solution under a modified condition based on the literature[31,32,41]. Typically, the ligand solution was prepared by dissolving 200 mg Ica (nIm, aIm) and 8 mg CTAB in 40 mL methanol/DMF (V/V = 3/1) at 60 °C for 3 h. 50 mg freshly-prepared Zn-ZIF-8@Pt@Zn-ZIF-8 was dispersed in 20 mL methanol, and then the prepared ligand solution was slowly added to the Zn-ZIFs dispersion under stirring conditions (1 mL/min). The ligand exchange reaction proceeded in a preheated oscillator at 50 °C for 2 days. After the reaction, the container was taken out of the oven and cooled to room temperature. The powder was centrifuged at $6000 \times g$ for 5 min and washed with methanol 3 times. Repeat this procedure two more times, and the Zn-ZIFs@Pt@Zn-ZIFs homologs were obtained. The solid was further soaked with fresh methanol (20 mL) for 24 h[31].

For Zn-ZIFs@Pt@Zn-ZIFs (CH2OH) (denoted as Zn-ZIFs@Pt@Zn-ZIFs (OH)), dried Zn-ZIFs@Pt@Zn-ZIFs (CHO) (50 mg) and $NaBH_4$ (75.6 mg) were suspended in methanol (20 mL), and then refluxed for 24 h. The reaction mixture was centrifuged at $6000 \times g$ for 5 min and the solid was washed 3 times with fresh methanol (20 mL). The solid was further soaked with fresh methanol (20 mL) for 24 h[32].

To quantify the rate of ligand exchange/reduction, the powder samples were dissolved in $D_2SO_4/D_2O$ and analyzed by solution $^1H$ NMR.

## Synthesis of SiO$_2$/Pt

$SiO_2$ sphere was prepared according to the Stöber procedure reported previously with some modifications[42]. 50 mg $SiO_2$ spheres were dispersed in 18 mL 10% HCl solution. The mixture was refluxed at 90 °C for 8 h under continuous stirring. The solid was collected by centrifugation and washed with $H_2O$. Subsequently, the solid was redispersed in 10 mL ethanol, and then 4 mL Pt NPs solution (1.2 mM) was added drop by drop under stirring. The mixture was stirred at room temperature for 2 h. Then the solid was centrifuged at $6000 \times g$ and washed with ethanol several times.

## In situ characterization

In situ Zn K-edge X-ray absorption spectra (XAS) were collected at the 1W1B beamline of the Beijing Synchrotron Radiation Facility (BSRF) equipped with a Si (111) double crystal monochromator by transmission mode. The storage ring of BSRF runs at 2.5 GeV with a maximum electron current of about 250 mA. The absorption edge of standard Zn foils is employed to calibrate the X-ray energy. For in situ $H_2$-flowing tests, the samples were stuffed in a polyimide tube and subsequently placed on a homemade heat plate. The X-ray absorption fine structure (XAFS) raw data were background-subtracted, normalized, and Fourier-transformed by the standard procedures with the ATHENA program. The extended X-ray absorption fine structure (EXAFS) curve fitting was performed with the ARTEMIS program. The in situ combined-XRD tests were conducted at the 1W1B beamline of BSRF with an incident X-ray energy of 9459 eV. The 2D diffraction images were collected by a 2D detector (DECTRIS Pilatus 3X, 100K-A, Rigaku) and visualized by ALBULA software package (Rigaku), and converted into 1D XRD patterns by using Dioptas software[43], with the default calibration procedure. In situ XPS was collected by Thermo Fischer, ESCALAB 250Xi XPS system under hydrogen atmosphere.

## Catalytic hydrogenation of cyclooctene (hydrogen spillover model reaction)

In a typical procedure, each sample (Zn-ZIFs@Pt@Zn-ZIFs homolog) containing the same amount of Pt NPs (0.004 mmol) was dispersed in 0.5 mL cyclooctene. Subsequently, the solution was transferred into a Teflon-lined stainless-steel autoclave, the autoclave was purged with $H_2$ 4 times, and the final $H_2$ pressure of the autoclave was set at the desired MPa. During the catalytic process, the reaction solution was magnetically stirred at the speed of 600 rpm at the desired temperature for the desired time. After that, the catalysts were separated by centrifugation, and washed with ethanol for reusing; while the obtained reaction solution was filtered through a filter membrane (0.22 μm), and then was analyzed by gas chromatography (GC, Shimadzu, GC-2010 Plus, column: Rtx-5, 30 m × 0.25 mm × 0.25 μm).

## Catalytic selective hydrogenation of 5-chloroquinolines catalyzed by various catalysts

In a typical procedure, each catalyst containing the same amount of Pt NPs (0.002 mmol) was dispersed in 2 mL isopropanol solution, and then 81.8 mg 5-chloroquinoline (0.5 mmol) was added to the above solution. Subsequently, the solution was transferred into a Teflon-lined stainless-steel autoclave, the autoclave was purged with H2 for 4 times, and the final H2 pressure of the autoclave was set at 2 MPa. During the catalytic process, the reaction solution was magnetically stirred at the speed of 600 rpm at 100 °C for 120 min. After that, the catalysts were separated by centrifugation, and washed with ethanol for reusing; while the obtained reaction solution was filtered through a filter membrane (0.22 μm), and then was analyzed by gas chromatography (GC, Shimadzu, GC-2010 Plus, column: Rtx-5, 30 m × 0.25 mm × 0.25 μm).

## Theoretical calculation

Our density functional theory (DFT) calculations[44,45] were carried out in the Vienna ab initio simulation package (VASP) based on the plane-wave basis sets with the projector augmented-wave method[46,47]. The exchange-correlation potential was treated by using a generalized gradient approximation (GGA) with the Perdew-Burke-Ernzerhof (PBE) parametrization[48]. The energy cutoff was set to be 520 eV. The Brillouin-zone integration was sampled with a Γ-centered Monkhorst-Pack mesh of $1 \times 1 \times 1$[49]. The structures were fully relaxed until the maximum force on each atom was less than 0.01 eV per Å, and the energy convergent standard was $10^{-5}$ eV. A Gaussian smearing with a width of 0.05 eV for the occupation of the electronic level was used. The climbing-image nudged elastic band (CI-NEB) method[50] as implemented in VASP code was used to determine the diffusion energy barrier and the minimum energy pathways for H evolution in ZIF. The intermediate images were relaxed until the forces were smaller than 0.02 eV per Å.

For implicit solvation calculations, we used VASPsol[51] that incorporates solvation into VASP within a self-consistent continuum model.

## Data availability

Additional data related to this work are available from the corresponding authors upon request.

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

## Acknowledgements

This work was supported by Strategic Priority Research Program of Chinese Academy of Sciences (grant numbers: XDB36000000, Z.Y.T.), National Key R&D Program of China (grant numbers: 2021YFA1200302 and 2022YFA1205400, Z.Y.T.) and National Natural Science Foundation of China (grant numbers: 92356304 and 92056204, Z.Y.T. grant numbers: 22203021, X.J.B.). We highly appreciate Prof. Jing Zhang and Dr. Pengfei An from the 1W1B beamline of the Beijing Synchrotron Radiation Facility (BSRF) for their support in synchrotron radiation XAS-XRD tests.

## Author contributions

Z.T. supervised the entire project. X.B. performed the synthesis, characterization, and catalysis experiments. C.Y. performed the in situ XAS-XRD experiments. X.B. wrote the manuscript. Z.T. edited the manuscript.

## Competing interests

The authors declare no competing interests.
