## [Peer review file · Nature Communications]

Enabling long-distance hydrogen spillover in nonreducible metal-organic frameworks for catalytic reactionREVIEWER COMMENTS

Reviewer #1 (Remarks to the Author):

In this manuscript, the authors report their recent investigation on determination of hydrogen travel distance of Zn-ZIF-8 cubic crystals via hydrogenation of cyclooctene on their external surfaces. The sandwich-like MOFs@Pt@MOFs structure is well suited for such an investigation, which allows the authors to ensure an accurate measurement of migrating distance of hydrogen atoms. In particular, the authors also identified the roles of included water and functionalization of ligands (into hydrogen acceptors) during the hydrogen transportation; this part of comparative findings provides useful information on the nature of mobility of hydrogen in nonreducible MOFs such as ZnZIF-8 in the current case. Overall speaking, the work has been nicely presented, and the conclusion has been carefully derived with a substantial set of supporting data in Supplementary Information. Thus, the manuscript can be recommended for its publication in NC after the following minor revisions.

- (1) Did the authors try other sizes of Pt nanoparticles? Is it possible for the authors to include the relationship between "spillover intensity" and "particle size" into the following statement: "The spillover intensity generally increases with thinner shell thickness, higher reaction temperature and longer reaction time."?
- (2) Figure Caption 2: Please indicate the same amount of Pt NPs in the catalysts used in the reactions.
- (3) Figure Caption 4: Please indicate the same amount of Pt NPs in the catalysts used in the reactions.
- (4) Some inconsistency in terminology should be corrected such as: ZIF-8@Pt@Zn-ZIF-8 (H₂O), where "ZIF-8" should be changed to "Zn-ZIF-8".

Reviewer #2 (Remarks to the Author):

This paper should be made revision before final consideration.

- 1) What is the evidence that hydrogenation reaction occurs using spillover hydrogen atom? It is possible that hydrogenation was occurred on Pt or border surface.
- 2) Isotope D atom should be used to understand mechanism.
- 3) What kind of surface functional group of MOF was used for hydrogen spillover. At that time where electron was captured.

Responses to Reviewers' and Editor's Comments

For Reviewer #1

Recommendation: Publish in NC after minor revisions.

General Comment: In this manuscript, the authors report their recent investigation on determination of hydrogen travel distance of Zn-ZIF-8 cubic crystals via hydrogenation of cyclooctene on their external surfaces. The sandwich-like MOFs@Pt@MOFs structure is well suited for such an investigation, which allows the authors to ensure an accurate measurement of migrating distance of hydrogen atoms. In particular, the authors also identified the roles of included water and functionalization of ligands (into hydrogen acceptors) during the hydrogen transportation; this part of comparative findings provides useful information on the nature of mobility of hydrogen in nonreducible MOFs such as ZnZIF-8 in the current case. Overall speaking, the work has been nicely presented, and the conclusion has been carefully derived with a substantial set of supporting data in Supplementary Information. Thus, the manuscript can be recommended for its publication in NC after the following minor revisions.

Our general response: We truly appreciate the reviewer's encouraging and insightful comments. All of the comments raised by the reviewer have been taken into account and answered in detail, which greatly improves the quality of our work.

Comment 1: Did the authors try other sizes of Pt nanoparticles? Is it possible for the authors to include the relationship between "spillover intensity" and "particle size" into the following statement: "The spillover intensity generally increases with thinner shell thickness, higher reaction temperature and longer reaction time."?

Our response: Thanks a lot for the reviewer's insightful suggestion. According to the reviewer's kind suggestion, we have synthesized larger size Pt nanoparticles (~5 nm) (below **Fig. R1**) to construct sandwich-like MOFs@Pt@MOFs catalysts and tested their spillover hydrogenation intensity. As shown in the **Fig. R2** and **Fig. R3**, the sandwich catalysts with large-sized Pt nanoparticles (Zn-ZIF-8@Pt(5 nm)@Zn-ZIF-8 (H₂O) and Zn-ZIFs@Pt(5 nm)@Zn-ZIFs (CHO)) exhibit nearly the same spillover decay factor (slopes in **Fig. R2** and **Fig. R3**) as the previously-used ones (Zn-ZIF-

8@Pt(3 nm)@Zn-ZIF-8 (H₂O) and Zn-ZIFs@Pt(3 nm)@Zn-ZIFs (CHO)) but lower hydrogenation activity, meaning that particle size only affects hydrogen activation independent of spillover intensity.

Fig. R1. TEM image of Pt nanoparticles and their size distribution.

Fig. R2. Linear relationship between spillover intensity and spillover distance of Zn-ZIF-8@Pt@Zn-ZIF-8 (H₂O) with different Pt nanoparticles size (80-minute reaction). All reactions were performed using catalysts with the same amount of Pt nanoparticles.

Fig. R2. Linear relationship between spillover intensity and spillover distance of Zn-ZIFs@Pt@Zn-ZIFs (CHO) (80-minute reaction). All reactions were performed using catalysts with the same amount of Pt nanoparticles.

Our revision: Following the reviewer's constructive suggestion, we have added the corresponding description: "In addition, the sandwich catalyst with large-sized Pt NPs (ca. 5 nm) exhibit nearly the same spillover decay factor as the above-used one but lower hydrogenation activity, meaning that particle size only affects hydrogen activation independent of spillover intensity (Supplementary Figs. 49-51)." (please see the yellow highlighted part on page 9 in the revised manuscript and **Supplementary Figs. 49-51** on page S58-S60 in the revised Supplementary Information.).

Comment 2: Figure Caption 2: Please indicate the same amount of Pt NPs in the catalysts used in the reactions.

Our response: Thanks a lot for the reviewer's careful reading and kind reminding. The amount of catalyst for catalytic reaction has been described in **Method** part and indicated in Figure Caption.

Our revision: Following the reviewer's kind suggestion, we have added the corresponding description: "All reactions were performed using catalysts with the same amount of Pt NPs." (Please see the yellow highlighted part on page 31 in the revised manuscript)

Comment 3: Figure Caption 4: Please indicate the same amount of Pt NPs in the catalysts used in the reactions.

Our response: Thanks a lot for the reviewer's careful reading. The amount of catalyst for catalytic reaction is described in **Method** part and indicated in Figure Caption.

Our revision: Following the reviewer's kind suggestion, we have added the corresponding description: "In a typical procedure, each catalyst containing the same amount of Pt NPs (0.002 mmol) was dispersed in 2 mL isopropanol solution, and then 81.8 mg 5-chloroquinolines (0.5 mmol) were added into the above solution." "All reactions were performed using catalysts with the same amount of Pt NPs." (Please see the yellow highlighted part on page 20 and page 34 in the revised manuscript).

Comment 4: Some inconsistency in terminology should be corrected such as: ZIF-8@Pt@Zn-ZIF-8 (H₂O), where “ZIF-8” should be changed to “Zn-ZIF-8”.

Our response: Thanks a lot for the reviewer’s careful reading and kind reminding. We have thoroughly revised and unified the terminology in the revised manuscript.

Our revision: Following the reviewer’s kind comment, we have corrected ZIF-8@Pt@Zn-ZIF-8 (H₂O) to Zn-ZIF-8@Pt@Zn-ZIF-8 (H₂O) (please see the yellow highlighted part on page 9 in the revised manuscript).

For Reviewer #2

Recommendation: Consideration in NC after revision.

General Comment: This paper should be made revision before final consideration.

Our general response: We truly appreciate the reviewer's critical comments. All of the comments raised by the reviewer have been taken into account and answered in detail, which greatly improves the quality of our work.

Comment 1: What is the evidence that hydrogenation reaction occurs using spillover hydrogen atom? It is possible that hydrogenation was occurred on Pt or border surface.

Our response: We highly appreciate the reviewer's critical comments. As illustrated in below **Fig. R4**, in the conventional MOFs supported metal nanoparticles (MOFs/metal NPs) catalysts, metal nanoparticles are randomly distributed in porous MOFs structure or near MOFs surface. So, both organic substrates and H₂ can diffuse onto the metal nanoparticle surface in the course of the catalytic reaction. This allows the reactions to occur not only on the MOFs surface but also on the metal nanoparticle surface, leading to difficulty in decoupling the contributions from both pathways.

Fig. R4. Hydrogenation pathways of cyclooctene on MOFs/metal NPs catalyst.

In this work, we design sandwich-like MOFs@Pt@MOFs catalysts, where metal catalysts are fully located inside with a uniform and tunable MOFs shell. It is noted that

Zn-ZIF-8 is characteristic of a very small aperture window (0.34 nm), allowing selective diffusion of H₂ (0.29 nm) rather than large-sized molecule like cyclooctene (0.6 nm). In this case, only small H₂ can access Pt nanoparticles but larger organic reactant cannot. So, all the catalytic activities are directly attributed to the reaction between spillover hydrogen and organic molecules on the external surface of MOFs matrix (**Fig. R5**, also Figure 3a in the revised manuscript).

Fig. R5. Hydrogenation pathway of cyclooctene on MOFs@Pt@MOFs.

To further prove that cyclooctene cannot pass through the Zn-ZIF-8 pore, comparative adsorption experiments were performed. First, Zn-ZIF-8 and Zn-ZIFs (CHO) were thoroughly dried and activated in a vacuum at 120°C, and then soaked in 1-hexene solution. After continuous stirring overnight, the samples were centrifuged with alcohol twice and dried naturally. The dried powders were dispersed in deuterium DMSO/D₂SO₄ and their ¹H NMR spectra were determined. As shown in **Fig. R6a** (also Supplementary Fig. 38a in the revised Supplementary Information), there exists a certain amount of 1-hexene, indicating the adsorption capability of Zn-ZIFs for small molecule 1-hexene. In sharp contrast, for larger size cyclooctene, no cyclooctene molecules are detected in Zn-ZIFs after the same treatment (**Fig. R6b**, also Supplementary Fig. 38b in the revised Supplementary Information), proving that cyclooctene could not enter the pore of Zn-ZIFs.

Fig. R6. (a) ^1H NMR spectra of 1-hexene, Zn-ZIF-8 after adsorption of 1-hexene and Zn-ZIFs (CHO) after adsorption of 1-hexene. (b) ^1H NMR spectra of cyclooctene, Zn-ZIF-8 after adsorption of cyclooctene and Zn-ZIFs (CHO) after adsorption of cyclooctene.

Comment 2: Isotope D atom should be used to understand mechanism.

Our response: Thanks a lot for the reviewers' insightful comment. Following the reviewer's kind suggestion, in order to understand the process of hydrogen spillover, isotope deuterium labeling experiments have been adopted to track the trajectories of water and hydrogen by detecting the probe molecule (cyclooctene) mass spectrometry. As shown in **Fig. R7** (the top line), the hydrogenation and deuteration of cyclooctene are hardly detected in Zn-ZIF-8(D_2O)- D_2 system, indicating that H-D exchange on probe cyclooctene is difficult to occur under the mild condition. At the same time, complete spillover hydrogenation and spillover deuteration can be clearly discerned in Zn-ZIF-8@Pt@Zn-ZIF-8 (H_2O)- H_2 system (the second line from the top) and Zn-ZIF-8@Pt@Zn-ZIF-8 (D_2O)- D_2 system (the third line from the top), respectively. Interestingly, whether in Zn-ZIF-8@Pt@Zn-ZIF-8 (D_2O)- H_2 system (the fourth line from the top) or in Zn-ZIF-8@Pt@Zn-ZIF-8 (H_2O)- D_2 system (the fifth line from the top), cyclooctene is hydrogenated to cyclooctane, $[\text{D}_1]$ -cyclooctane and $[\text{D}_2]$ -cyclooctane. The above analysis demonstrates that that H_2 splitting occurs on Pt nanoparticles, and the activated hydrogen atoms diffuse across MOFs structure by the water-assist path accompanied by exchange with water.

In addition, as for Zn-ZIFs@Pt@Zn-ZIFs (CHO)- D_2 system (the bottom line), the

product of [D₂]-cyclooctane clarifies the migration of D atoms across MOFs containing CHO functional groups.

Fig. R7. Deuterium labeling experiments. Analysis of reaction products in different systems.

Our revision: Following the reviewer’s kind suggestion, we have added the corresponding description: “In order to elucidate the process of hydrogen spillover, isotope deuterium labeling experiments were adopted to track the trajectories of water and hydrogen by detecting the probe molecule (cyclooctene) mass spectrometry (Supplementary Fig. 57). As shown in Supplementary Fig. 58, the hydrogenation and

deuteration of cyclooctene are hardly detected in Zn-ZIF-8(D₂O)-D₂ system, indicating that H-D exchange on probe cyclooctene is difficult to occur under the mild condition. At the same time, complete spillover hydrogenation and spillover deuteration can be clearly discerned in Zn-ZIF-8@Pt@Zn-ZIF-8 (H₂O)-H₂ system and Zn-ZIF-8@Pt@Zn-ZIF-8 (D₂O)-D₂ system, respectively. Interestingly, whether in Zn-ZIF-8@Pt@Zn-ZIF-8 (D₂O)-H₂ system or in Zn-ZIF-8@Pt@Zn-ZIF-8 (H₂O)-D₂ system, cyclooctene is hydrogenated to cyclooctane, [D₁]-cyclooctane and [D₂]-cyclooctane. The above analysis demonstrates that H₂ splitting occurs on Pt NPs, and the activated hydrogen atoms diffuse across MOFs structure by the water-assist path accompanied by exchange with water. In addition, as for Zn-ZIFs@Pt@Zn-ZIFs (CHO)-D₂ system, the product of [D₂]-cyclooctane clarifies the migration of D atoms across MOFs containing CHO functional groups.” (Please see the yellow highlighted part on page 10 in the revised manuscript and **Supplementary Figs. 57-58** on page S66-S68 in the revised Supplementary Information).

Comment 3: What kind of surface functional group of MOF was used for hydrogen spillover. At that time where electron was captured.

Our response: We are grateful for the reviewer's critical comments. Zeolitic imidazolate framework-8 (Zn-ZIF-8) is composed of zinc ions [Zn²⁺] as inorganic nodes connected to 4 organic ligands of 2-methylimidazole (**Fig. R8**. Blue structure).

Fig. R8. Crystal structure of ZIF-CHO (pink) and ZIF-CH₃ (blue) with the same sodalite (SOD) topologies.

Hydrogen spillover in pure Zn-ZIF-8 is inefficient, but it can be largely promoted by changing the methyl group of ligand 2-methylimidazole to other functional groups (such as aldehyde group, hydroxyl group and amino group) while keeping the same topological structure (**Fig. R8**). So, we propose the hydrogen spillover pathway based on ligand functional groups. Using aldehyde group as a typical example, as shown in *in situ* H₂-XPS spectra (**Fig. R9**) (also Supplementary Fig. 70 in the revised Supplementary Information), the corresponding O 1s peak (CHO) clearly shifts to a higher energy under hydrogen atmosphere, indicating that the O species of the aldehyde group is the binding site of the H atom. In general, the activated hydrogen atom produced on Pt nanoparticle is first adsorbed at the oxygen site of the aldehyde group near Pt nanoparticle, and then continuously transferred to the aldehyde group (oxygen site) of the well-structured ligand functional group. Therefore, the process is a transfer of atomic hydrogen, and it does not involve the capture of electron. Furthermore, the calculated low transfer energy barrier (0.47 eV) of the transfer of hydrogen atom between neighboring oxygen sites confirms the functional group-spillover pathway (**Fig. R10**) (also Fig. 5b in the revised manuscript).

Fig. R9. *In situ* XPS spectra (O 1s) of Zn-ZIFs@Pt@Zn-ZIFs (CHO) at as-prepared and spillover state.

Fig. R10. Hydrogen spillover on aldehyde-Zn-ZIFs (steps O1-TS₁₋₂-O2 of left model): $E_{\text{act}} = 0.47$ eV.

REVIEWERS' COMMENTS

Reviewer #1 (Remarks to the Author):

In this revised manuscript, the authors have essentially addressed the concerns of this reviewer. The quality and testing scope of manuscript in its present form have been further improved. Therefore, the manuscript can be recommended for its publication in Nature Communications if other reviewers are also satisfied with the authors' additional experimental work and revisions.

Reviewer #2 (Remarks to the Author):

because moderate modification and comment were made, this paper can be accepted.